# Efficacy and safety of prazequantel for the treatment of *Schistosoma mansoni* infection across different transmission settings in Amhara Regional State, northwest Ethiopia

Getaneh Alemu[1]*, Arancha Amor[2], Endalkachew Nibret[3,4], Abaineh Munshea[3,4], Melaku Anegagrie[2]

1 Department of Medical Laboratory Science, College of Medicine and Health Sciences, Bahir Dar University, Bahir Dar, Ethiopia, 2 Mundo Sano Foundation and Institute of Health Carlos III, Madrid, Spain, 3 Biology Department, Science College, Bahir Dar University, Bahir Dar, Ethiopia, 4 Health Biotechnology Division, Institute of Biotechnology (IoB), Bahir Dar University, Bahir Dar, Ethiopia

* ggetanehmlt@gmail.com

**Data Availability Statement:** All relevant data are within the manuscript and its Supporting information files.

## Abstract

### Background

*Schistosoma mansoni* and *S. haematobium* infections have been public health problems in Ethiopia, *S. mansoni* being more prevalent. To reduce the burden of schistosomiasis, a national school-based prazequantel (PZQ) mass drug administration (MDA) program has been implemented since November 2015. Nevertheless, *S. mansoni* infection is still a major public health problem throughout the country. Reduced efficacy of PZQ is reported by a few studies in Ethiopia, but adequate data in different geographical settings is lacking. Hence, this study aimed to assess the efficacy and safety of PZQ for the treatment of *S. mansoni* infection across different transmission settings in Amhara Regional State, northwest Ethiopia.

### Methods

A school-based single-arm prospective cohort study was conducted from February to June, 2023 among 130 *S. mansoni*-infected school-aged children (SAC). Forty-two, 37, and 51 *S. mansoni*-infected SAC were recruited from purposely selected schools located in low, moderate, and high transmission districts, respectively. School-aged children who were tested positive both by Kato Katz (KK) using stool samples and by the point of care circulating cathodic antigen (POC-CCA) test using urine samples at baseline were treated with a standard dose of PZQ and followed for 21 days for the occurrence of adverse events. After three weeks post-treatment, stool and urine samples were re-tested using KK and POC-CCA. Then the cure rate (CR), egg reduction rate (ERR), and treatment-associated adverse events were determined. The data were analyzed using SPSS version 21.

### Results

Out of the total 130 study participants, 110 completed the follow-up. The CR and ERR of PZQ treatment were 88.2% (95%CI: 82.7–93.6) and 93.5% (95%CI: 85.4–98.5),

**Funding:** The author(s) received no specific funding for this work.

**Competing interests:** The authors have declared that no competing interests exist.

**Abbreviations:** ARHB, Amhara Regional Health Bureau; CCA, Circulating Cathodic Antigen; CR, Cure Rate; EPG, Eggs Per Gram of stool; ERR, Egg Reduction Rate; FMoH, Federal Ministry of Health; KK, Kato Katz; MDA, Mass Drug Administration; POC-CCA, Point of Care Circulating Cathodic Antigen; PZQ, Prazequantel; SAC, School-aged children; SCH, Schistosomiasis; SSA, sub-Saharan Africa; STHs, Soil Transmitted Helminths; WASH, Water Sanitation and Hygiene; WHO, World Health Organization.

respectively, by KK. The CR of PZQ based on the POC-CCA test was 70.9% (95%CI: 62.7–79.1) and 75.5% (95%CI: 67.3–83.6) depending on whether the interpretation of 'trace' results was made as positive or negative, respectively. After treatment on the 21[st] day, 78 and 83 participants tested negative both by KK and POC-CCA, with respective interpretations of 'trace' POC-CCA test results as positive or negative. The CR in low, moderate and high transmission settings was 91.7%, 91.2% and 82.5%, respectively (p = 0.377) when evaluated by KK. The CR among SAC with a light infection at baseline (95.7%) by KK was higher than that of moderate (81.5%) and heavy (64.3%) infections ($\chi^2$ = 12.53, p = 0.002). Twenty-six (23.6%) participants manifested at least one adverse event. Eleven (10.0%), eight (7.3%), six (5.5%), and three (2.7%) participants complained about abdominal pain, nausea, headache, and anorexia, respectively. All adverse events were mild, needing no intervention. Occurrence of adverse events was slightly higher in high endemic areas (32.5%) than moderate (23.5%) and low endemic areas (p = 0.279).

## Conclusions

A single dose of 40 mg/kg PZQ was efficacious and safe for the treatment of *S. mansoni* infection when it was evaluated by the KK test, but a lower efficacy was recorded when it was evaluated by the POC-CCA test. However, the POC-CCA test's specificity, clearance time of CCA from urine after treatment, and interpretation of weakly reactive (trace) test results need further research.

## Background

Schistosomiasis (SCH) is a water-borne disease caused by blood-dwelling flukes of the genus *Schistosoma* [1]. It is endemic in 70 developing countries where 700 million people are at risk of infection. More than 200 million people are infected globally, with more than 90% of the infections occurring in sub-Saharan Africa (SSA) [1–3]. It causes a loss of about 3.3 million disability-adjusted life years [3] and up to 280,000 deaths annually in the SSA [4]. *Schistosoma mansoni* and *S. haematobium* are the predominant species affecting humans globally, while *S. japonicum* is restricted to the Asian continent [5, 6]. Other human-infecting species: *S. intercalatum*, *S. mekongi*, *S. guineensis*, and *S. malayensis*, have limited and focal geographical distributions [7, 8].

Schistosomiasis is a major public health problem in Ethiopia, where about 53.3 million people are at risk of infection [9]. *Schistosoma mansoni* and *S. haematobium* are found in Ethiopia. *Schistosoma mansoni* is widespread throughout the country, whereas *S. haematobium* is restricted to the lowland borders of the country [10, 11]. The snail intermediate host of *S. mansoni*, *Biomphalaria* spp., adapts best to altitudes ranging from 1300 to 2000 meters above sea level, which covers a large landmass of Ethiopia. The presence of a suitable snail intermediate host as well as a conducive physical environment made *S. mansoni* prevalent throughout the country [12, 13]. Despite the patchy distribution of SCH in Amhara Regional State, the regional prevalence of *S. mansoni* infection ranges from 17.5% to 41.1% based on recent systematic reviews and meta-analyses data [14–17].

The Federal Ministry of Health (FMoH) of Ethiopia has been implementing a national SCH control program to halt the transmission by the year 2025 [9] so as to achieve the WHO's 2030 target of eliminating SCH in all endemic countries [18]. The FMoH has identified and

implemented five strategies to accomplish this goal: mass drug administration (MDA) using prazequantel (PZQ); facility-based case diagnosis and treatment; snail vector control; water, sanitation, and hygiene (WASH); and social and behavioral change communication [9]. In Ethiopia and other endemic countries, MDA has been the cornerstone of control programs because it is relatively easily applied and is supported by drug donors [19].

Despite the implementation of the MDA program for many years, the prevalence of SCH did not drop to an acceptable level in Ethiopia [9]. The first report of reduced PZQ efficacy (egg reduction rate (ERR) <80%) in Ethiopia was published in 2012 by Erko and his colleagues based on a study from Wondo genet, southern Ethiopia [20]. Recently, one study reporting reduced PZQ efficacy [21] and another study reporting doubtful PZQ efficacy (ERR of 80% —<90%) [22] were published in 2020 and 2022, respectively, from Amhara region. This is alarming and demands continuous monitoring of PZQ efficacy in order to inform policy-makers of control programs about the effectiveness of the ongoing MDA in reducing disease transmission. In this way, the World Health Organization (WHO) recommends monitoring anti-helminthic drug efficacy if the drug is used for MDA for four or more years, or if the disease prevalence did not drop to an acceptable level despite the implementation of MDA [23]. Soil-transmitted helminthes (STHs), especially *Ascaris lumbricoides* and hookworms, are co-endemic with *S. mansoni* in Ethiopia, and hence PZQ and albendazole/mebendazole are co-administered to school-aged children during MDA campaigns [9]. However, PZQ efficacy in schistosoma mono-infection and STH- co-infected children is not well addressed in Ethiopia. Moreover, the KK method that is being used for the assessment of PZQ efficacy is not sensitive enough for detecting light infections [17, 24, 25], and as a result, overestimation of CR is highly likely. In light of this fact, WHO suggested the use of urine point of care circulating cathodic antigen (POC-CCA) test along with KK in endemic countries for monitoring and evaluation of SCH control programs [26]. Circulating cathodic antigen (CCA) is one of the major schisto-somal gut-associated glycoproteins regurgitated by the live adult parasites, and majority is secreted in urine. Hence, the detection of CCA in urine is suggestive for the presence of live worms. Despite the fact that all *Schistosoma* species produce CCA, the highest concentrations of the antigen are detected in *S. mansoni* infections, and therefore the POC-CCA test is partic-ularly useful to diagnose *S. mansoni* infections. As shown in the Schisto POC-CCA® test kit manufacturer information sheet, CCA rapidly declines after successful treatment, and a posi-tive test result usually becomes negative within 2 to 3 weeks after treatment [27, 28]. However, there is no adequate data about the performance of POC-CCA in assessing cure after PZQ treatment. Hence, the present study aimed to assess PZQ efficacy and safety for the treatment of *S. mansoni* infection and to compare the capacity of POC-CCA and KK for assessing cure in different transmission settings.

## Methods and materials

### Study design, area, and period

From February to June 2023, a school-based single-arm prospective cohort study was con-ducted among School-aged children (SAC) attending selected primary schools in different transmission settings of intestinal schistosomiasis in the Amhara Regional State, northwest Ethiopia. Data collection areas were stratified into three based on low (1% to <10%), moderate (>10% to <50%), and high (>50%) *Schistosoma* transmission settings [9]. The majority of the population in the region lives in rural areas, where agriculture accounts for 39% of domestic products. Only 20% and 69% of households have access to improved latrine and drinking water sources, respectively. Moreover, large natural and artificial open water sources, includ-ing Lake Tana, the Blue Nile River, Koga Dam, and Tana Beles sugar cane irrigation, are found

in and around the present study area, making the area suitable for SCH. Amhara Regional Health Bureau (ARHB), in collaboration with FMoH, conducted a regional SCH mapping in 2015, updated in 2020 [9]. Among 81 districts mapped in northwest Ethiopia, 49, 17, 11, and 4 districts were found to be non-endemic (<1% prevalence), low (1%-<10%), moderate (10%-<50%), and high (≥50%) endemic for *S. mansoni* infections, respectively (ARHB, unpublished report).

## Sample size calculation

The WHO protocol recommends a minimum of 50 *S. mansoni*-positive participants be enrolled in a PZQ efficacy study [23]. Hence, we calculated the sample size to be screened to get 50 positive children from each of the low, moderate, and high transmission settings using *S. mansoni* prevalence of 29.9% [29], 44.8% [30], and 89.9% [31] from previous studies conducted using KK in low, moderate, and high endemic settings, respectively. Just to avoid confusion about having 29.9% in low-endemic area, the schistosomiasis endemicity level in Ethiopia is declared at a district level. However, due to its focal distribution nature, some localities within a low-endemic district might have higher than the upper limit of low transmission (1%—< 10%). Assuming an 80% compliance rate, the sample size to be screened was calculated by the formula: sample size = 50/(0.8*prevalence). Replacing prevalences of 29.9%, 44.8%, and 89.9%, the sample sizes for low, moderate, and high transmission districts were 209, 140, and 70, respectively.

## Sampling technique

The study participants were SAC aged 6–14 who attended selected primary schools. Schools were selected purposively to represent low, moderate, and high transmissions of SCH in districts based on the SCH endemicity map [9]. Accordingly, two schools from a low transmission setting (Andasa Primary School from Bahir Dar Zuria District, and Tach Qorata Primary School from Dera District), two schools from a moderate transmission setting (Chagni 01 Primary School and Chagni 03 Primary School from Chagni town), and one school from a high transmission setting (Ewqet Amba Primary School from Tach Armachiho District) were selected. The sample size for each endemicity level was proportionally allocated to each school and to each grade level based on the number of children attending each selected school and grade. Finally, participants for initial screening were selected from each class by a systematic random sampling technique using class rosters as a sampling frame. The first participant was enrolled on 16 February 2023 and the data collection end date was 27 June 2023.

## Data collection and laboratory diagnosis methods

**Questionnaire data collection.** Data on socio-demographic characteristics and clinical signs and symptoms were collected by trained nurses using structured questionnaires. Questionnaires were administered to SC through a face-to-face interview. The questionnaires were translated from English to Amharic (the local language) before administration.

**Sample collection and processing.** Stool and urine samples were collected from each SAC to detect *S. mansoni* infection. After teaching about the collection process, two wide-mouthed cups (one for stool and the other for urine) were given to children to collect about 5g of stool and 3ml of random midstream urine samples at school. Urine samples were used for the detection of CCA using the POC-CCA test strip. Stool samples were transported in a triple package at room temperature to nearby health centers and processed by KK on the day of collection. The KK and POC-CCA tests were conducted by trained medical laboratory technologists.

**Urine POC-CCA test.**   A Schisto POC-CCA® test (Rapid Medical Diagnostics; Cape Town, South Africa; Batch Number: 220902098; Expire date: 2024/09) was performed from a urine sample at schools immediately after collection following the manufacturer's procedures. In brief, two drops of urine (100 μl) were added to the sample port well of the test cassette placed in a horizontal position. After applying the urine, the CCA antigen that may be present in the sample binds to the labeled monoclonal antibody immobilized in the sample membrane. The solution then runs over the strip, where the antigen-antibody complex attaches to the monoclonal antibody immobilized at the test line to develop a pink line. Results were read just after 20 minutes and interpreted as positive if a pink line developed at the test line and negative if a pink line did not develop. The presence of a pink control line was used to make sure the test worked correctly.

**Stool KK test.**   A fresh stool sample was processed by the KK technique at nearby health center laboratories. Approximately 2–3 gram of fresh stool was sieved through a fine wire mesh of size 200 μm and the stool filtered off over the mesh was scraped using a plastic spatula. A template with a hole, which is assumed to sample 41.7 mg of feces, was placed at the center of a microscope slide, and the hole was filled with the sieved fecal material. The template was then gently removed. A cellophane cover slip, pre-soaked in glycerol-malachite green solution for 24 hours, was placed over the fecal material and pressed to have a uniform smear. Slides were examined for the detection of *S. mansoni* ova. The number of eggs per gram of stool (EPG) was calculated by multiplying the number of eggs per slide by 24. The EPG was calculated from duplicate KK slide readings separately, and the average was taken as the final EPG [32, 33]. The KK smear was also used to detect the ova of STHs [33].

## Inclusion and exclusion criteria

Schoolchildren aged 6–14 who were positive for *S. mansoni* infection both by KK and POC-CCA, regardless of their STH infection status, were enrolled for assessment of PZQ efficacy. Schoolchildren who lived in the study area at least for six months and attended a regular primary school to participate in the study were included. Moreover, participants were included in the study after obtaining assent from the children and consent from their parents or caregivers. Schoolchildren who had diarrhea and took any anti-helminthic drugs within six months prior to data collection were excluded from the study. Children who were co-infected with intestinal helminthes other than *S. mansoni* and STHs were also excluded.

## Evaluation of prazequantel efficacy

Evaluation of PZQ efficacy was conducted following the WHO protocol [23]. Children were instructed to eat breakfast before taking PZQ treatment. A single dose of 40 mg/Kg PZQ (brand name: biltricide, expired date: 24/11/2023, manufacturer: Bayer pharmaceuticals) purchased from a private pharmacy in Bahir Dar, Ethiopia, was administered to each eligible child directly observed during drug uptake and were followed for four hours for vomiting, if any. Children who vomited within four hours after drug administration were withdrawn from the follow-up. Trained health extension workers regularly followed students from day 1 to day 7 on a daily basis to record any adverse events. The severity of adverse events was graded based on the Common Terminology Criteria for Adverse Events version 5.0 [34]. After three weeks, stool and urine samples were collected from each participant and examined by KK and POC-CCA.

## Study outcomes and definitions

The primary study outcome in the present study was PZQ efficacy (cure rate (CR) and ERR at three weeks post-treatment). Cure rate was calculated from the results of the KK and POC-CCA tests separately, while ERR was calculated based on EPG reported by the KK pre- and post-treatment. The CR by KK was defined as the proportion of treated children who were egg-positive at baseline but became egg-negative at week three after treatment. The CR by POC-CCA was defined as the proportion of treated children who were POC-CCA test positive at baseline but became POC-CCA test negative at week three after treatment. Cure rate and ERR were calculated based on the WHO protocol as follows [23].

$$CR(\%) = \frac{\text{(Number of negative participants who tested positive at baseline)}}{\text{Number of positive participants at baseline}} \times 100$$

$$ERR(\%) = 1 - \frac{\text{(Arithmetic mean egg counts at follow-up)}}{\text{Arithmetic mean egg counts at baseline}} \times 100$$

A point estimate of ERR of $\geq$90% was considered satisfactory; between 80% and 90%, and <80% were considered doubtful and reduced efficacy, respectively [23].

The secondary outcome of the present study was incidence and type of adverse events that occurred after PZQ treatment. An adverse event was defined as any sign or symptom that was not reported before PZQ treatment but occurred during the follow-up period. For this purpose, study participants were interviewed for any signs and symptoms (nausea, anorexia, headache, cough, fever, vomiting, stomach pain, rash, dizziness, confusion, difficulty to breath, diarrhea, itching, and any other symptoms). Treated children were prospectively monitored for seven days on a daily basis for the occurrence of any adverse events.

## Data quality assurance

Training was given to data collectors (medical laboratory technologists, health extension workers, nurses, and teachers). In KK and POC-CCA test procedures, standard operating procedures and the manufacturer's instructions were strictly followed. The expiration date of all KK reagents (glycerol and malachite green) and POC-CCA test kits was checked before use. All KK smears and POC-CCA tests were independently read by two medical laboratory technologists, and discordant readings were read by a third senior medical laboratory technologist. The presence of the control line was checked in all POC-CCA tests. The PZQ package was inspected for any physical damage, and its expiration date was double-checked during purchasing and just before treatment. The KK set, POC-CCA test kits, and PZQ were stored appropriately as per the manufacturer's instructions until use.

## Data analysis

Data were entered and analyzed in SPSS software version 21 (IBM SPSS Corp. Chicago, USA). A descriptive statistic was used to analyze the socio-demographic characteristics, clinical history, and side effects. Pearson's Chi-square test was run to assess an association between potential risk factors with CR and adverse events. The non-parametric Mann-Whitney U test was used to compare the mean EPG among males vs. females, aged 6–9 vs. 10–14, STH-infected vs. non-infected, while the Kruskal-Wallis H test was used to compare the mean EPG among participants with light, moderate, and heavy intensity of infections.

### Ethics approval and consent to participate

The present study was approved by the Institutional Review Board of Bahir Dar University, Science College (Ref: PRCSVD/514/2015). A support letter was obtained from the Amhara Public Health Institute. Permission letter was obtained from selected districts health and education offices and selected primary schools. Informed written consent was obtained from parents/caregivers of participating children, and assent was obtained from children. All participants tested positive for *S. mansoni* and STHs were treated with the standard doses of PZQ and Albendazole, respectively. Participants with other intestinal parasite infections were linked to nearby health institutions for treatment. Participants who were not cured (positive at follow-up) were re-treated with a single dose of 40mg/kg PZQ.

## Results

### Socio-demographic characteristics of study participants

A total of 167, 112, and 64 participants were screened in low, moderate, and high transmission settings. The response rate in the low and moderate transmission settings was low, probably due to low awareness about the public health importance of *S. mansoni* infection. Among the total participants screened, 130 were positive for *S. mansoni* by both diagnostic methods and hence enrolled in the PZQ efficacy study. Of the 130 participants, twenty were excluded from the data analysis because eight students were absent on the treatment date; four refused to take the drug; one vomited within four hours after taking the drug; and seven were absent on the follow-up dates. Hence, 110 participants were included in the final analysis. The majority (88.2%) of participants were aged 10–14 and 56.4% were male participants (Table 1).

### Cure rate

Study participants recruitment and flow chart with CR stratified by diagnostic methods is presented in Fig 1. Among 110 participants who completed the follow-up, 97 were tested negative by stool KK three weeks post-treatment, giving a CR of 88.2% (95%CI: 82.7–93.6). When

**Table 1. Socio-demographic characteristics and STH co-infection among school-aged children participated in prazequantel efficacy testing in northwest Ethiopia, February to June 2023 (N = 110).**

| Variable | Catagory | Number (%) | STH co-infection | |
| --- | --- | --- | --- | --- |
| | | | Hookworms N(%) | *Ascaris lumbricoides* N(%) |
| Age group (in years) | 6–9 | 13(11.8) | 0 | 0 |
| | 10–14 | 97(88.2) | 11(11.3) | 2(2.1) |
| Sex | Male | 62(56.4) | 7(11.3) | 1(1.6) |
| | Female | 48(43.6) | 4(8.3) | 1(2.1) |
| School Grade | 1–4 | 72(65.5) | 6(8.3) | 2(2.8) |
| | 5–8 | 38(34.5) | 5(13.2) | 0 |
| District | Bahir Dar Zuria | 17(15.5) | 0 | 0 |
| | Chagni town | 34(30.9) | 0 | 1(2.9) |
| | Dera | 19(17.3) | 10(52.6) | 1(5.3) |
| | Tach Armachiho | 40(36.4) | 1(2.5) | 0 |
| School | Andasa Primary School | 17(15.5) | 0 | 0 |
| | Chagni 01 Primary School | 16(14.5) | 0 | 0 |
| | Chagni 03 Primary School | 18(16.4) | 0 | 1(5.6) |
| | Tach Qorata Primary School | 19(17.3) | 10(52.6) | 1(5.3) |
| | Euket Amba Primary School | 40(36.4) | 1(2.5) | 0 |

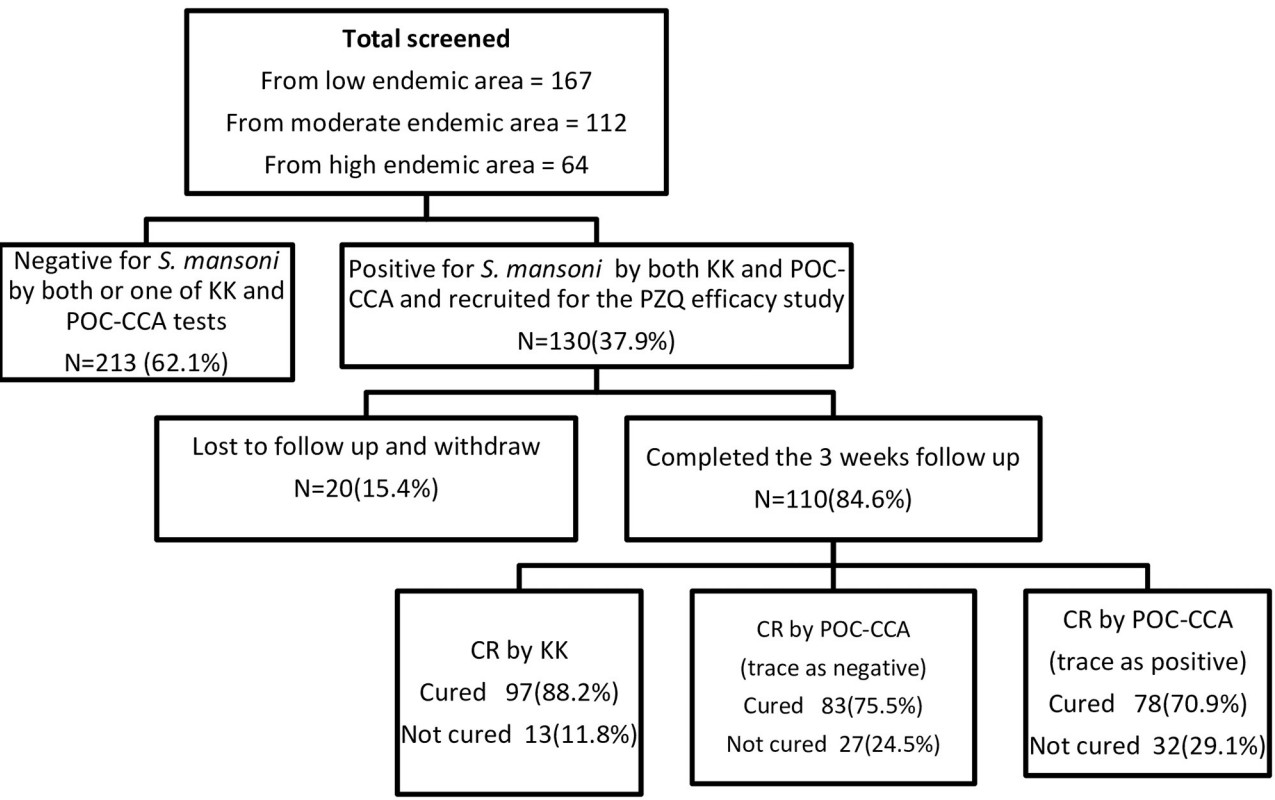

**Fig 1. Study flow chart and observed cure rates stratified by diagnostic methods.**

'trace' results were considered positive in POC-CCA test, the CR was found to be 70.9% (95% CI: 62.7–79.1) which was lower than the CR evaluated by KK ($\chi^2$ = 35.93, p < 0.001). When 'trace' results were considered negative, the CR was 75.5% (95%CI: 67.3–83.6). This was significantly lower compared to the CR that was calculated based on KK results ($\chi^2$ = 45.32, p < 0.001) (Table 2).

A total of 69 (62.7%), 27 (24.5%), and 14 (12.7%) participants had light, moderate and heavy *S. mansoni* infections, respectively, at baseline. Light, moderate and heavy infections were reported from all the three (low, moderate and high) transmission settings but the

**Table 2. Comparison of cure rate of PZQ as diagnosed by KK and POC-CCA methods among school-aged children in northwest Ethiopia, February to June 2023 (N = 110).**

| | | Cured (POC-CCA test; trace result as positive) | | | $\chi^2$ | P-value |
|---|---|---|---|---|---|---|
| | | Yes | No | Total | 35.934 | <0.001 |
| Cured (KK test) | Yes | 78 | 19 | 97 (88.2%) | | |
| | No | 0 | 13 | 13 | | |
| | Total | 78 (70.9%) | 32 | 110 | | |
| | | Cured (POC-CCA test; trace result as negative) | | | 45.319 | <0.001 |
| | | Yes | No | Total | | |
| Cured (KK test) | Yes | 83 | 14 | 97 (88.2) | | |
| | No | 0 | 13 | 13 | | |
| | Total | 83 (75.5%) | 27 | 110 | | |

**Table 3. Prazequantel cure rate and associated risk factors among school-aged children in northwest Ethiopia, February to June 2023 (N = 110).**

| Variable | Category | By KK | | | | By POC-CCA (trace as negative) | | | | By POC-CCA (trace as positive) | | | |
|---|---|---|---|---|---|---|---|---|---|---|---|---|---|
| | | Cured(%) | Not cured (%) | $\chi^2$ | p-value | Cured (%) | Not cured (%) | $\chi^2$ | p-value | Cured (%) | Not cured (%) | $\chi^2$ | p-value |
| Age group (in years) | 6–9 | 11(84.6) | 2(15.4) | 0.18 | 0.671 | 9(69.2) | 4(30.8) | 0.31 | 0.579 | 8(61.5) | 5(38.5) | 0.63 | 0.428 |
| | 10–14 | 86(88.7) | 11(11.3) | | | 74(76.3) | 23(23.7) | | | 70(72.2) | 27(27.8) | | |
| Sex | Male | 54(87.1) | 8(12.9) | 0.16 | 0.689 | 45(72.6) | 17(27.4) | 0.63 | 0.426 | 42(67.7) | 20(32.3) | 0.69 | 0.406 |
| | Female | 43(89.6) | 5(10.4) | | | 38(79.2) | 10(20.8) | | | 36(75) | 12(25) | | |
| Grade level | 1–4 | 65(90.3) | 7(9.7) | 0.88 | 0.349 | 57(79.2) | 15(20.8) | 1.55 | 0.213 | 52(72.2) | 20(27.8) | 0.17 | 0.676 |
| | 5–8 | 32(84.2) | 6(15.8) | | | 26(68.4) | 12(31.6) | | | 26(68.4) | 12(31.6) | | |
| District | Bahir Dar Zuria | 16(94.1) | 1(5.9) | 2.14 | 0.544 | 14(82.4) | 3(17.6) | 0.54 | 0.910 | 14(82.4) | 3(17.6) | 2.58 | 0.461 |
| | Chagni town | 31(91.2) | 3(8.8) | | | 25(73.5) | 9(26.5) | | | 21(61.8) | 13(38.2) | | |
| | Dera | 17(89.5) | 2(10.5) | | | 14(73.7) | 5(26.3) | | | 14(73.7) | 5(26.3) | | |
| | Tach Armachiho | 33(82.5) | 7(17.5) | | | 30(75) | 10(25) | | | 29(72.5) | 11(27.5) | | |
| School | Andasa | 16(94.1) | 1(5.9) | 2.53 | 0.639 | 14(82.4) | 3(17.6) | 2.53 | 0.640 | 14(82.4) | 3(17.6) | 4.61 | 0.330 |
| | Chagni 01 | 14(87.5) | 2(12.5) | | | 10(62.5) | 6(37.5) | | | 8(50) | 8(50) | | |
| | Chagni 03 | 17(94.4) | 1(5.6) | | | 15(83.3) | 3(16.7) | | | 13(72.2) | 5(27.8) | | |
| | Tach Qorata | 17(89.5) | 2(10.5) | | | 30(75) | 10(25) | | | 29(72.5) | 11(27.5) | | |
| | Euket Amba | 33(82.5) | 7(17.5) | | | 14(73.7) | 5(26.3) | | | 14(73.7) | 5(26.3) | | |
| SCH endemicity* | Low | 33 (91.7) | 3 (8.3) | 1.95 | 0.377 | 28(77.8) | 8(22.2) | 0.18 | 0.915 | 28(77.8) | 8(22.2) | 2.25 | 0.325 |
| | Moderate | 31(91.2) | 3 (8.8) | | | 25(73.5) | 9(26.5) | | | 21(61.8) | 13(38.2) | | |
| | High | 33(82.5) | 7 (17.5) | | | 30(75) | 10(25) | | | 29(72.5) | 11(27.5) | | |
| Intensity of infection | Light | 66(95.7) | 3(4.3) | 12.53 | 0.002 | 55(79.7) | 14(20.3) | 3.24 | 0.198 | 54(78.3) | 15(21.7) | 4.87 | 0.088 |
| | Moderate | 22(81.5) | 5(18.5) | | | 20(74.1) | 7(25.9) | | | 16(59.3) | 11(40.7) | | |
| | Heavy | 9(64.3) | 5(35.7) | | | 8(57.1) | 6(42.9) | | | 8(57.1) | 6(42.9) | | |
| STH co-infection | Yes | 12(92.3) | 1(7.7) | 0.24 | 0.624 | 10(76.9) | 3(23.1) | 0.02 | 0.896 | 10(76.9) | 3(23.1) | 0.26 | 0.611 |
| | No | 85(87.6) | 12(12.4) | | | 73(75.3) | 24(24.5) | | | 68(70.1) | 29(29.9) | | |
| Overall | | 97(88.2) | 13(11.8) | | | 83(75.5) | 27(24.5) | | | 78(70.9) | 32(29.1) | | |

*endemicity of data collection districts based on the Ethiopian FMoH national SCH mapping result [9]

majority of heavy infections (11 out of 14) were from a high transmission setting as data presented in S1 File. There was a significant association between the baseline intensity of infection and CR ($\chi^2$ = 12.53, p = 0.002). Children with a light infection at baseline had the highest CR (95.7%), followed by moderate (81.5%), and heavy (64.3%) infections. However, CR was not associated with SCH endemicity of data collection districts (p > 0.05). There was no significant association between age, sex, data collection district, STH co-infections and CR (Table 3).

## Egg reduction rate

The mean baseline EPG among study participants was 163.9 ± 205.3 (mean ± standard deviation, SD). The minimum and maximum EPG at baseline were 12 and 1104, giving a range of 1092. This large range makes the standard deviation from the mean EPG large. The median EPG was 72 while the most frequent EPG was 24 at baseline. The mean, median, mode, minimum, maximum, and range of EPG both at baseline and follow-up are presented in S1 File. The mean EPG varied by age group and sex, but the difference was not significant (p > 0.05). The mean EPG among STH-co-infected children (42.5 ± 24.3) significantly differed (p = 0.002) from the mean EPG among STH-negative children (180.1 ± 213.4). The mean EPG

**Table 4. Adverse events observed among school-aged children after prazequantel treatment across associated risk factors in northwest Ethiopia, February to June 2023 (N = 110).**

| Variable | Category | Number of children assessed n | Adverse event n(%) | $\chi^2$ | P-value |
|---|---|---|---|---|---|
| Age group (in years) | 6–9 | 13 | 6(46.2) | 3.72 | 0.054 |
| | 10–14 | 97 | 21(21.6) | | |
| Sex | Male | 62 | 11(17.7) | 3.55 | 0.060 |
| | Female | 48 | 16(33.3) | | |
| Transmission setting of data collection sites | Low | 36 | 6(16.7) | 2.59 | 0.274 |
| | Moderate | 34 | 8(23.5) | | |
| | Heavy | 40 | 13(32.5) | | |
| Intensity of infection | Light | 69 | 14(20.3) | 5.29 | 0.071 |
| | Moderate | 27 | 11(40.7) | | |
| | Heavy | 14 | 2(14.3)) | | |
| STH co-infection | Yes | 13 | 0(0.0) | 4.80 | 0.029 |
| | No | 97 | 27(27.8) | | |
| Number of PZQ tablets taken | <3 tablets | 96 | 23(24.0) | 0.14 | 0.743 |
| | ≥3 tablets | 14 | 4(28.6) | | |
| Overall | | 110 | 27(24.5) | | |

at follow-up was 7.2 ± 23.8, providing an overall ERR of 93.5% (95%CI: 85.4–98.5), as presented in S2 File.

## Adverse events after prazequantel treatment

Out of 110 study participants, 27 adverse events occurred in 26 participants. The adverse events reported were abdominal pain (11/110, 10.0%), nausea (8/110, 7.3%), headache (6/110, 5.5%), and anorexia (3/110, 2.7%). Only one adverse event occurred in each of the 26 participants, while two adverse events occurred in one participant. The majority of adverse events (26/27, 96.3%) occurred on the first day of treatment and were resolved by day 2. One adverse event (headache) occurred two days after treatment and stayed for a half day before resolving. All adverse events were mild. Adverse events were more common among children with no co-infection with STHs as compared to those with STH co-infection (p = 0.029). Participants' age, sex, and baseline *S. mansoni* infection intensity were not significantly associated with the occurrence of adverse events (Table 4).

## Discussion

The CR of PZQ as evaluated by the KK method (88.2%, 95%CI: 81.8–93.6) was high, revealing that a single dose of 40 mg/kg PZQ is effective in treating *S. mansoni* infections. The CR was nearly equal to the upper limit of the standard CR of 60%-90% set by the WHO for single-dose PZQ treatment [35]. The present finding is in line with earlier results [22, 36–42]. The CR in the present study was higher than previous reports from Ethiopia [43] and other SSA countries [20, 44–46]. On the contrary, the CR in the present study was lower than previous reports from Ethiopia [47] and Tanzania [24]. Variations in the study population, baseline infection intensity, and diagnostic methods as well as parasite, host, and drug associated factors might contribute for the difference in CR [48].

The lower CR using the POC-CCA test as compared to the CR measured by KK (p<0.001) implies that PZQ is not as efficacious as it was thought before. The present PZQ CR using POC-CCA is similar to previous reports [25, 26] but higher than the findings of other studies

[24, 44, 49]. The previous studies measured CRs six weeks and above after treatment, possibly leading to re-infections and thereby contributing to the lower CR. The POC-CCA test could be an important alternative for assessment of cure. This is in line with the WHO's suggestion to use POC-CCA along with KK for mapping and monitoring the success of MDA in *S. mansoni* transmission settings [26]. However, until the specificity of the technique is well determined, we should be cautious in the interpretation of POC-CCA test results, as we do not know how long it takes for schistotoma antigens to clear from the urine, and we can face false positive results. Urinary tract infections and haematuria are also thought to cause false-positive results. Therefore, the specificity, actual CCA clearance time after treatment, correlation of test results with infection intensity, PZQ mechanism of action to the adult female worms (whether reducing fecundity or killing), and doubt whether juvenile worms are releasing detectable amounts of CCA or not should be known before recommending POC-CCA alone for the assessment of cure [24].

As measured by the KK method, participants with light infection at baseline had the highest CR, while the lowest CR was reported among participants with heavy infection (95.7% vs. 64.3%; p = 0.002). Similar findings were reported from southern Ethiopia [36]. In cases of moderate and heavy infections, few adults might escape from the action of PZQ, contributing to a reduced CR as compared to light infections. Moreover, the insensitivity of KK, especially to detect light infections, might contribute to false-negative results and hence a higher CR among participants with light infection at baseline. However, CR was not associated with the endemicity level of data collection districts. The actual baseline intensity of infection for each participant determines the CR better than the SCH endemicity of study areas.

Recent studies reported that lower age groups had lower CR, probably due to a lower dose of PZQ they took as the treatment was on height [40, 50]. Similarly, in our study, we found that participants aged 6–9 had a lower CR as compared to those aged 10–14, but the difference was not significant (84.6% vs. 88.7%, p = 0.671). In the present study, a small number of participants (only 13) were enrolled within the lower age group, which made it difficult to provide strong justification.

As presented in S2 File, STH co-infection was significantly associated with baseline *S. mansoni* infection intensity (p = 0.002) and all STH co-infected children had a light *S. mansoni* infection. The majority of STH infections were caused by hookworms (11 out of 13), which feed on blood similar to *S. mansoni*. Competition for nutrients might reduce the fecundity of *S. mansoni* and hence lower the EPG among hookworm-co-infected participants compared to *S. mansoni* mono-infections. However, STH co-infection was not associated with CR (p > 0.05) which was in line with a previous study in southern Ethiopia [36]. The number of co-infected participants in the present study was small, which makes it difficult to draw a definitive conclusion.

In the present study, the ERR (93.5%) as measured by duplicate KK three weeks post-treatment was above the WHO minimum threshold (90%), indicating that PZQ is still satisfactory for the treatment of *S. mansoni* infection [23]. This result is in line with previous reports in Ethiopia [36, 37, 41, 43, 47] and elsewhere in Africa [24, 46]. Doubtful [22, 45] and reduced [20, 39, 44] ERR were reported by other previous studies. The difference might be due to variations in the baseline infection intensity, diagnostic methods used, and length of the follow-up period [20, 22, 39, 45, 46]. The prolonged follow-up time in previous studies might affect the actual ERR due to re-infection and maturation of parasites that were juvenile at the time of treatment. Moderate and heavily infected children had a higher ERR as compared to children with light-intensity infections. This result is similar to previous findings [36]. It has been known that single-dose PZQ reduces female worm fecundity, thereby decreasing egg output after treatment [51].

In the present study, occurrence of treatment-associated adverse events (27/110, 24.5%) was higher as compared to a previous report of 17% [30], but it was lower than the incidence of adverse events from other studies [20, 38, 50]. Abdominal pain and nausea were the most common adverse events, which was in line with previous reports [36, 52]. In our study, all adverse events were mild and transient, needing no intervention. Rigorous assessment of pre-existing baseline clinical data, ensuring whether children have eaten breakfast just before treatment, accurate drug dosage, counseling about the drug intake procedure, and the nature of the drug might contribute to the low incidence of adverse events observed among SC in the present study. Despite only mild and transient adverse events were reported in the present study, integrating safety surveillance with the ongoing MDA enables early detection of any emerging adverse events and associated factors. Participants without STH co-infection were more likely to develop adverse events (p = 0.029). It is difficult to justify this because STH-co-infected children were small in number (13) and all co-infected participants had only a light *S. mansoni* infection. Since adverse events are believed to occur because of worm-drug interaction, light infections might cause decreased or no adverse events.

The present study has the following limitations: the sample size was small, which makes it difficult to compare PZQ efficacy across low, moderate, and high endemicity settings. The small sample size decreases the strength of our conclusions too. Whether the reported adverse events were due to drug therapy or any other cause was not assessed.

## Conclusions

Prazequantel was efficacious for the treatment of *S. mansoni* infection when it was evaluated by the KK test. A single dose of 40 mg/kg PZQ caused mild adverse events, so it is safe for the treatment of *S. mansoni* infection. Therefore, we recommend continual use of a single dose 40 mg/kg PZQ as a first-line treatment for *S. mansoni* infection. However, the efficacy was recorded as low when it was evaluated by a more sensitive POC-CCA test, which is an indication of KK overestimating PZQ efficacy. The urine POC-CCA test could be a potential alternative. However, POC-CCA test specificity, clearance time of CCA from urine after treatment, and interpretation of 'trace' results need further research before recommending this test alone for diagnosis and assessment of cure. Continuous, large-scale country-wide studies are recommended to early detect reduced efficacy and to draw a strong conclusion about the effect of transmission setting on PZQ efficacy.

## Supporting information

**S1 File. Mean, median, mode, minimum, maximum and range of *S. mansoni* fecal egg counts among schoolchildren in northwest Ethiopia, February to June 2023 (N = 110).** (DOCX)

**S2 File. Mean fecal egg count and egg reduction rate among schoolchildren in northwest Ethiopia, February to June 2023 (N = 110).** (DOCX)

## Acknowledgments

We thank the Mundo Sano Foundation and Institute of Health Carlos III for providing POC-CCA test kits; Bahir Dar University Science College for ethical approval of the study; Amhara Regional Health Bureau for providing support letter; and study participants for their volunteer participation in the study; Tadesse Hailu for reviewing the manuscript.

## Author Contributions

**Conceptualization:** Getaneh Alemu, Arancha Amor.

**Data curation:** Getaneh Alemu, Arancha Amor, Endalkachew Nibret, Abaineh Munshea, Melaku Anegagrie.

**Formal analysis:** Getaneh Alemu, Arancha Amor, Endalkachew Nibret.

**Funding acquisition:** Melaku Anegagrie.

**Investigation:** Getaneh Alemu, Arancha Amor, Melaku Anegagrie.

**Methodology:** Getaneh Alemu, Arancha Amor, Endalkachew Nibret, Abaineh Munshea, Melaku Anegagrie.

**Project administration:** Getaneh Alemu, Arancha Amor, Abaineh Munshea.

**Resources:** Arancha Amor.

**Software:** Getaneh Alemu, Arancha Amor, Endalkachew Nibret.

**Supervision:** Arancha Amor, Endalkachew Nibret, Abaineh Munshea, Melaku Anegagrie.

**Validation:** Abaineh Munshea, Melaku Anegagrie.

**Visualization:** Getaneh Alemu, Arancha Amor, Endalkachew Nibret, Abaineh Munshea, Melaku Anegagrie.

**Writing – original draft:** Getaneh Alemu.

**Writing – review & editing:** Arancha Amor, Endalkachew Nibret, Abaineh Munshea, Melaku Anegagrie.

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
