## [Decision Letter · Decision Letter 0]

7 Dec 2023

PONE-D-23-33684Efficacy and safety of praziquantel for the treatment of Schistosoma mansoni infection across different transmission settings in Amhara Regional State, northwest EthiopiaPLOS ONE

Dear Dr. Abebe,

Thank you for submitting your manuscript to PLOS ONE. After careful consideration, we feel that it has merit but does not fully meet PLOS ONE’s publication criteria as it currently stands. Therefore, we invite you to submit a revised version of the manuscript that addresses the points raised during the review process.

We look forward to receiving your revised manuscript.

Kind regards,

Clement Ameh Yaro, Ph.D

Academic Editor

PLOS ONE

Journal Requirements:

3. Thank you for stating the following in your Competing Interests section: "NO authors have competing interests".

Reviewers' comments:

Reviewer's Responses to Questions

**Comments to the Author**

1. Is the manuscript technically sound, and do the data support the conclusions?

Reviewer #1: Partly

Reviewer #2: Yes

Reviewer #3: Yes

2. Has the statistical analysis been performed appropriately and rigorously? 

Reviewer #1: No

Reviewer #2: Yes

Reviewer #3: Yes

3. Have the authors made all data underlying the findings in their manuscript fully available?

Reviewer #1: Yes

Reviewer #2: Yes

Reviewer #3: Yes

4. Is the manuscript presented in an intelligible fashion and written in standard English?

Reviewer #1: No

Reviewer #2: Yes

Reviewer #3: Yes

5. Review Comments to the Author

Reviewer #1: The authors present a very interesting study in which they investigated the efficacy and safety of praziquantel for S. mansoni in an endemic setting in Ethiopia. The authors indicate the objective of the study which is to investigate the efficacy of praziquantel in different transmission settings. This sounds very interesting and I was looking forward to reading the results. However, I was surprised that the results do not mention anything about the efficacy of treatment in the different transmission settings - as stated in the objective of the study. Instead, results are reported according to different (demographic) variables. In my opinion, table 3 and table 4 should show the CR by KK and POC-CCA specified according to transmission setting (low/moderate/high) as this was the main objective. Table 3 and 4 in their current state are rather complex and do not provide the evidence to answer the main question. Suggestion to include them as a supplement.

The discussion is very elaborate, and includes various statements not directly relevant for this study. For example, comparison of cure rates with other studies (lines 271-294) - this is often not possible due to various factors, i.e. the diagnostic methods used as well as the time frame after treatment used to measure efficacy, but also the setting in which the other studies were performed (different countries, low/moderate/high endemic, study population). Since the authors find a relative high CR based on KK, I do not see the need to mention in detail other studies finding lower cure rates. Of course, it is good to put the data of this study in perspective, but then more focused and leaving out unnecessary details.

Also, the discussion of resistance (lines 278-284) is more informative and does not contribute to the interpretation of study findings, ie can resistance be the explanation that the CR is different between KK and POC-CCA?

In the conclusion the authors state that the POC-CCA test specificity, clearance time of CCA and interpretation of traces need further research BEFORE recommending this test for diagnosis and treatment monitoring. Even though I fully agree that in some specific settings (ie. urinary tract infections, pregnancy, young children) there are issues with specificity, and that the interpretation of traces can be a challenge, I find the conclusion the authors make - that these issues need to be investigated before the test can be used - quite strong. In particular since the POC-CCA test is already being recommended by WHO to be used in addition to or as an alternative for KK in S. mansoni settings. Perhaps the authors can reflect on this in their discussion/conclusion, in particular since they mention it briefly in the introduction (see also comment below)

Overall, I miss any conclusion about the efficacy of treatment in different transmission settings and my suggestion would be to re-analyse the data according to this objective and re-write the results and discussion accordingly. I think this makes the study unique and would be very interesting to report. The current manuscript does not report anything new, it is generally known that PZQ is safe and efficacious (depending on the diagnostic method used).

Specific comments / questions

1. Please have a native English speaker read the manuscript and check for grammatical errors.

2. Please use references appropriately:

line 96: ref 17 is not correct, this paper does not mention anything about efficacy of PZQ - please check and include the correct reference

line 98: ref 25 incorrect, this report does not mention anything about POC-CCA - please check and include the correct reference

lines 100-105: please add references for POC-CCA statements. There are many publications describing studies using the POC-CCA.

2. Methods - Urine POC-CCA test: please add batch number and expiration date, this is important so the study can be compared to other studies that have used the same batch of POC-CCA.

3. Methods - Efficacy of PZQ: treated children were counseled to avoid activities related to infection. How did you monitor this? Especially since follow-up is 3 weeks after treatment. How to determine what chidlren did in the mean time?

4. Methods - Data analysis: here STH infection is mentioned as a variable. Nothing is mentioned about this in the diagnostics section. How did you determine the presence of STH? Which STHs? Are they common in the study area? Did you also treat in case a person was infected with any STH?

5. Methods - Ethics: What happened to those individuals who remained Schistosoma positive at follow-up? Did they receive another treatment?

6. Results: Suggestion to include a flow chart as a first figure to better explain / visualize the study flow and to have a clear picture of how many individuals were included in the study and analysis.

7. Results: Since you indicate you also measured STH infections, I would like to see the details of what you have found in terms of STH infections. For example in Table 1 or as a supplement.

Reviewer #2: Comment to the author

Dear the author, congratulations on the work. It has valuable and sound scientific relevance regardless of current programs for the prevention of transmission and elimination of the disease. Therefore, some comments should be understood and questions should be briefly answered.

1.The introduction part is good if there is any decreased efficacy in the study country in the previous study when and where was the first result reported?

2.The methodology needs to explore inclusion and exclusion criteria further.

3.What was your measurement for screening stunted children and nutritional status?

4.What were the criteria for recruiting STH co-infected participants and how were they managed?

5.The sample size seems too small, what sample size is needed to screen for low, moderate, and high transmission settings?

6.What would you do if the intended sample size on screen resulted in more negative results?

7.Before administrating the drug to the study participant, was it inspected for quality after being purchased from a private drug store?

8.The drug lacks information regarding its expiration date.

9.Regardless of the drug’s safety assessment, how did you distinguish between adverse events and side effects, clinical signs and symptoms of other co-infections?

10.What are the standard terminology criteria for adverse events of the study drug?

11.Can teachers and health extension workers determine adverse events?

12.What was your management of those participants who had vomiting during drug administration?

13.During the laboratory examination, how did you control personal visual effects during the POC-CCA test reading?

14.Quality control of laboratory results?

15.Lab professionals and laboratory are still not clear.

Result

16.Create a flowchart that screened, excluded for various reasons, lost to follow-up, and completed participants for better understanding for the reader.

17.What were your primary and secondary outcomes regarding the cure rate and safety assessment of the drug?

18.The results did not adequately describe adverse events, nor discuss when they happened and the duration of recovery.

19.What is your suggestion about the association between STH co-infection and the absence of adverse events?

20.Is the result of the POC-CCA test acceptable after 21 days post-administration of the drug as a cure rate if the drug fails to kill the antigen produced by juvenile parasites?

21.Insert if any limitation

22.The conclusion might be okay if you recommend continuous study at a large area and the country level.

Reviewer #3: Author : Tukwarlba et al.

Title : Efficacy and safety of praziquantel for the treatment of Schistosoma mansoni infection

across different transmission settings in Amhara Regional State, northwest Ethiopia

Journal : PLoS Negl Trop Dis

Reference: PONE-D-23-33684

General comment

The authors of the current study are working on a well-known issue. The data could be useful for Ethiopia. Here I have some comments and suggestion for improvement.

Title

= “Efficacy and safety of praziquantel for the treatment of Schistosoma mansoni infection across different transmission settings in Amhara Regional State, northwest Ethiopia” This title should be revised to consider the content of the MS. Note sure if this study was implemented in several transmission settings. What is clear is that the study was targeting SAC with different level of infection intensity (low, moderate and heavy)

Abstract

= “Factors such as reduced PZQ efficacy after … contributing to persistent transmission of the disease.” Regarding the context in Ethiopia, this could not the rationale behind this study. The authors should find good rationale supporting the current work

Introduction

= “More than 240 million people are infected globally” This statement is not anymore true

= “Reduced PZQ efficacy following multiple rounds of MDA might have contributed for the continued incidence and this has been evidenced by recent studies.” As mention above the rationale behind this study must be reconsidered.

Materials and methods

= “From February to June 2023, a school-based single-arm ….. different transmission settings of intestinal schistosomiasis” Which are these different settings? How those settings have been selected?

= “the calculated sample size was 635. For the PZQ efficacy study, we selected the above mentioned five schools to which a total of 283 participants were allocated.” What is the precise sample size of this work? How precisely this sample size has been estimated? Please, generate a new section entitled “Sample size calculation” and describe clearly how the sample size has been assessed.

= Please, define clearly your primary outcome and secondary outcome and show us how this has been considered in your sample size calculation.

= L151-L161 “Urine POC-CCA test” this part could be shorten avoiding speculation

= “Duplicate KK slides were prepared and examined from each sample and the average egg count was taken as the final EPG” This statement is not correct. Please, revise since 24 to get the EPG should multiply the average egg count from both slides.

=”Schoolchildren who lived in the study area at least for six months, attended a regular primary school, gave consent/assent to participate in the study were included.” SAC could not consent they are only assenting and their parents are consenting.

=”Evaluation of praziquantel efficacy” Please specify the brand of the praziquantel used in the current study

= L188 – L190 “Treated children were then counseled to avoid activities related to S. mansoni infection …. in a pot/jar for ≥24 hours before use).” Remove this part since praziquantel is not active against juvenile worms. So, whatever the precautions you are taking to avoid reinfection you could not clear the juvenile work in the children body. The follow up time set to three weeks has been set to limit the impact of reinfection on cure rate and ERR calculation.

= How many time the adverse events have been follow ?

= L214: “STH infected vs non-infected” This is the first time STH are mentioned. Nowhere, in the previous section something is said about STH.

= L214: “while Kruskal-Wallis H test was used to compare the mean EPG among participants with light, moderate and heavy intensity of infections” Why this comparison since the groups are already well defined as low, moderate and heavy infected participants?

= L234: Please write “Cure rate” instead of “Cure rate and associated risk factors”

Results

= Figure 1: should be revised to make it understandable including a figure caption

= Table 4: Please, could you provide confidence interval for the ERR

= Table 5 should be revised as follow:

Variable

CategoryNumber of children assessedAdverse event

n (%)

χ2

P-value

Discussion

= “The CR among SC with light infection (95.7%) by KK was higher than the CR among SC with moderate (81.5%) and heavy (64.3%) infections (χ2 = 12.53, p = 0.002)” One issue here is also the insensitivity of the KK in low infected people and need to be discussed.

= Please, make you discussion around your key findings by avoiding speculation

= Due to the limited sample size in the different study groups, the results of the current study should be interpreted with caution.

6. PLOS authors have the option to publish the peer review history of their article (what does this mean?). If published, this will include your full peer review and any attached files.

Reviewer #1: **Yes: **P.T. Hoekstra

Reviewer #2: No

Reviewer #3: **Yes: **Jean T. Coulibaly

---

## [Author Response · Author response to Decision Letter 0]

17 Dec 2023

Responses to Reviewers’ Comments and Questions

Dear editor and reviewers, thank you for your constructive comments which all are important inputs for the betterment of the manuscript. Below we tried to respond to all the comments/questions one by one; we also have incorporated all the corrections in the revised manuscript (shown as highlighted). 

Editor 

Response: we have revised the manuscript to meet the PLOS ONE’s style requirements

2. We suggest you thoroughly copyedit your manuscript for language usage, spelling, and grammar.

Response: we have made language editing thoroughly by ourselves and using Quillbot online editor available at; https://quillbot.com/grammar-check

3. Please complete your Competing Interests on the online submission form to state any Competing Interests.

Response: we have completed the form 

Response: because we were unable to get permission from the original copyright holder, we have removed the figure from the revised submission. 

Reviewer #1 

General comments

The authors present a very interesting study in which they investigated the efficacy and safety of praziquantel for S. mansoni in an endemic setting in Ethiopia. The authors indicate the objective of the study which is to investigate the efficacy of praziquantel in different transmission settings. This sounds very interesting and I was looking forward to reading the results. However, I was surprised that the results do not mention anything about the efficacy of treatment in the different transmission settings - as stated in the objective of the study. Instead, results are reported according to different (demographic) variables. In my opinion, table 3 and table 4 should show the CR by KK and POC-CCA specified according to transmission setting (low/moderate/high) as this was the main objective. Table 3 and 4 in their current state are rather complex and do not provide the evidence to answer the main question. Suggestion to include them as a supplement.

Response: we accepted this comment. We re-analyzed the CR by KK and POC-CCA according to transmission settings or endemicity of data collection districts and we incorporated the results in Table 3. We have also added in the ‘discussion’ section about CR across low, moderate and high transmission settings. We attached table 4 as supporting information, as you suggested. 

The discussion is very elaborate, and includes various statements not directly relevant for this study. For example, comparison of cure rates with other studies (lines 271-294) - this is often not possible due to various factors, i.e. the diagnostic methods used as well as the time frame after treatment used to measure efficacy, but also the setting in which the other studies were performed (different countries, low/moderate/high endemic, study population). Since the authors find a relative high CR based on KK, I do not see the need to mention in detail other studies finding lower cure rates. Of course, it is good to put the data of this study in perspective, but then more focused and leaving out unnecessary details.

Also, the discussion of resistance (lines 278-284) is more informative and does not contribute to the interpretation of study findings, ie can resistance be the explanation that the CR is different between KK and POC-CCA?

Response: we accepted the comment and we have made revisions. We have removed sentences explaining about drug resistance. We have also removed unnecessary comparisons with previous studies (please see the first paragraph of the discussion).

In the conclusion the authors state that the POC-CCA test specificity, clearance time of CCA and interpretation of traces need further research BEFORE recommending this test for diagnosis and treatment monitoring. Even though I fully agree that in some specific settings (ie. urinary tract infections, pregnancy, young children) there are issues with specificity, and that the interpretation of traces can be a challenge, I find the conclusion the authors make - that these issues need to be investigated before the test can be used - quite strong. In particular since the POC-CCA test is already being recommended by WHO to be used in addition to or as an alternative for KK in S. mansoni settings. Perhaps the authors can reflect on this in their discussion/conclusion, in particular since they mention it briefly in the introduction (see also comment below)

Response: we have added our reflections in the ‘discussion’ section of the revised manuscript

Overall, I miss any conclusion about the efficacy of treatment in different transmission settings and my suggestion would be to re-analyse the data according to this objective and re-write the results and discussion accordingly. I think this makes the study unique and would be very interesting to report. The current manuscript does not report anything new, it is generally known that PZQ is safe and efficacious (depending on the diagnostic method used).

Response: we have re-analyzed the data by transmission settings. However, the efficacy doesn’t show significant difference (shown in the revised Table 3) by transmission setting or endemicity level of data collection districts. Rather base line infection intensity of each participant determines the PZQ efficacy. We have presented this in the results, discussion and conclusion sections of the revised manuscript.

Specific comments / questions

1. Please have a native English speaker read the manuscript and check for grammatical errors.

Response: we have made a language editing 

2. Please use references appropriately:

line 96: ref 17 is not correct, this paper does not mention anything about efficacy of PZQ - please check and include the correct reference

Response: here we want to explain about the poor sensitivity of the KK method in general and we believe the reference is correct. Anyways, in the revised manuscript, we have cited two more references which provide information about the insensitivity of KK in monitoring PZQ efficacy. 

line 98: ref 25 incorrect, this report does not mention anything about POC-CCA - please check and include the correct reference

Response: sorry for this mistake. We have cited an appropriate reference in the revised manuscript

lines 100-105: please add references for POC-CCA statements. There are many publications describing studies using the POC-CCA.

Response: we have cited references in the revised manuscript.

2. Methods - Urine POC-CCA test: please add batch number and expiration date, this is important so the study can be compared to other studies that have used the same batch of POC-CCA.

Response: we have added the batch number and expiration date in the revised manuscript (Batch No. 220902098, expire date 2024/09)

3. Methods - Efficacy of PZQ: treated children were counseled to avoid activities related to infection. How did you monitor this? Especially since follow-up is 3 weeks after treatment. How to determine what chidlren did in the mean time?

Response: Trained health extension workers visited children every day while they attend school. They interviewed children about any health complain for the last 24hrs.Parents of participating children were informed to call to the health extension worker immediately if there was any health problem while children are at home. Health extension workers living in the same locality with study participants were employed for the follow up. Trained school teachers were also supporting the health extension workers.

4. Methods - Data analysis: here STH infection is mentioned as a variable. Nothing is mentioned about this in the diagnostics section. How did you determine the presence of STH? Which STHs? Are they common in the study area? Did you also treat in case a person was infected with any STH?

Response: We have added explanations about STHs in the ‘background’ and ‘methods (KK test)’ sections.

STHs (especially A. lumbricoides and hookworms) are co-endemic with S. mansoni in Ethiopia and hence PZQ and albendazole/mebendazole are co-administered to school-aged children during MDA campaigns. Hence we were interested to assess if there is a difference in CR and ERR among groups who were infected with S. mansoni alone (treated with PZQ) and groups who were co-infected (treated with PZQ plus Albendazole). We have treated STH infected participants with a standard dose of Albendazole.

5. Methods - Ethics: What happened to those individuals who remained Schistosoma positive at follow-up? Did they receive another treatment? 

Response: we re-treated them with a similar dose (single dose of 40mg/kg)

6. Results: Suggestion to include a flow chart as a first figure to better explain / visualize the study flow and to have a clear picture of how many individuals were included in the study and analysis.

Response: we have added a figure showing the participants recruitment and study flow in the revised manuscript

7. Results: Since you indicate you also measured STH infections, I would like to see the details of what you have found in terms of STH infections. For example in Table 1 or as a supplement.

Response: we accept the comment and we have incorporated STH findings in Table 1 of the revised manuscript

Reviewer #2 

Dear the author, congratulations on the work. It has valuable and sound scientific relevance regardless of current programs for the prevention of transmission and elimination of the disease. Therefore, some comments should be understood and questions should be briefly answered.

1.The introduction part is good if there is any decreased efficacy in the study country in the previous study when and where was the first result reported?

Response: we accept the comment and we have made revisions based on your comment (highlighted in the ‘Background’ section)

2.The methodology needs to explore inclusion and exclusion criteria further.

Response: we have revised the ‘inclusion and exclusion criteria’ section to exhaustively mention the eligibility criteria we have considered.

3. What was your measurement for screening stunted children and nutritional status?

Response: we have screened for nutritional status using body-mass-index (BMI) calculated from their height and weight. We didn’t assess stunting among participants. 

4. What were the criteria for recruiting STH co-infected participants and how were they managed?

Response: STHs (especially A. lumbricoides and hookworms) are co-endemic with S. mansoni in Ethiopia and hence PZQ and albendazole/mebendazole are co-administered to school-aged children during MDA campaigns. Hence we were interested to assess if there is a difference in CR and ERR among groups who were infected with S. mansoni alone (treated with PZQ) and groups who were co-infected (treated with PZQ plus albendazole). We have added explanations about STHs and inclusion criteria in the ‘background’, ‘methods’ and ‘results’ sections.

5. The sample size seems too small, what sample size is needed to screen for low, moderate, and high transmission settings? 

Response: we agree that if the sample size was larger, a more representative data would be generated but due to logistic issues and following the WHO (2013) protocol, we calculated the sample size to be screened to get 50 S. mansoni positive participants in each of low, moderate and high transmission settings considering a compliance rate of 80% (sample size calculation presented in more detail in the revised manuscript). But there was high refusal to give consent, loss to follow up, and withdrawal after enrolment that only 110 participants have complete data for analysis. We have mentioned the small sample size as a limitation in the revised manuscript. 

6. What would you do if the intended sample size on screen resulted in more negative results?

Response: we would have screened more children from selected or nearbye schools.

7. Before administrating the drug to the study participant, was it inspected for quality after being purchased from a private drug store?

Response: we only checked the expiry date, and inspected that the drug package and tablets were dry and had no physical damage.

8. The drug lacks information regarding its expiration date.

Response: expire date mentioned in the revised manuscript.

9. Regardless of the drug’s safety assessment, how did you distinguish between adverse events and side effects, clinical signs and symptoms of other co-infections?

Response: this was very challenging. We have added a sub-topic ‘study outcomes and definitions’ under the ‘Methods’ section to define the study variables and assumptions we considered. We carefully registered any clinical sign and symptoms (of any cause) before administering the drug. Then any new sign and symptoms developed after PZQ therapy (which was not observed before treatment) was considered as ‘adverse event’. 

We understood that the adverse events may or may not be related to the drug therapy. We mentioned as a limitation in the ‘discussion’ section of the revised manuscript.

10.What are the standard terminology criteria for adverse events of the study drug?

Response: it is a descriptive terminology which can be utilized for Adverse Event (AE) reporting. It is available at: https://ctep.cancer.gov/protocoldevelopment/electronic_applications/docs/ctcae_v5_quick_reference_5x7.pdf

11.Can teachers and health extension workers determine adverse events?

Response: Health extension workers can determine adverse events after getting training. Teachers were trained just to support health extension workers (corrected in the revised manuscript).

12.What was your management of those participants who had vomiting during drug administration?

Response: one child had vomited immediately after PZQ intake. We placed the child in a private ventilated area. She vomited only once and she didn’t loss significant amount of fluid. After evaluation by the nurse (one of the study team), the child was found needing no rehydration or any other treatment. She was taken to a near bye health center and stayed under follow up for one day. Then she was sent home.

13. During the laboratory examination, how did you control personal visual effects during the POC-CCA test reading?

Response: as stated in the ‘quality assurance’ section each POC-CCA test was read by two medical laboratory technologists independently. If there was discrepancy between the two, a third senior laboratory technologist (tie-breaker) read the result. 

14. Quality control of laboratory results?

Response: we have explained more about the quality control of laboratory results (highlighted in the ‘data quality assurance’ section). 

15.Lab professionals and laboratory are still not clear.

Response: we have replaced these terms with more appropriate ones in the revised manuscript (Lab professionals replaced by medical laboratory technologists, ‘laboratory’ replaced by KK and POC-CCA) as highlighted in the ‘sample collection and processing’ and ‘data quality assurance’ sections.

Result

16.Create a flowchart that screened, excluded for various reasons, lost to follow-up, and completed participants for better understanding for the reader.

Response: we have constructed a flow chart in the revised submission (please see Fig 2).

17.What were your primary and secondary outcomes regarding the cure rate and safety assessment of the drug?

Response: We have added a sub-topic ‘study outcomes and definitions’ under the ‘Methods’ section to define the primary (PZQ efficacy) and secondary (occurrence and type of adverse events) study variables and assumptions we considered.

18.The results did not adequately describe adverse events, nor discuss when they happened and the duration of recovery.

Response: we have made revision to present more about the adverse events (incidence, duration, severity) in the ‘results’ section. Revisions are also made in the ‘discussion’ section. 

19. What is your suggestion about the association between STH co-infection and the absence of adverse events?

Response: we have added more explanations about association of STH-co-infections with cure rate and occurrence of adverse events in the revised manuscript. Participants without STH co-infection were more likely to develop adverse events (p=0.029). It is difficult to justify this because STH co-infected children were small in number (13) and all co-infected participants had only light S. mansoni infection. Since adverse events are believed to occur because of worm-drug interaction, light infections cause decreased or no adverse events.

20.Is the result of the POC-CCA test acceptable after 21 days post-administration of the drug as a cure rate if the drug fails to kill the antigen produced by juvenile parasites?

Response: this is one area of debate to use POC-CCA alone for assessment of cure and we have raised this issue in the ‘discussion’ section. What age of the parasite does CCA starts to be released at detectable concentration is yet to be exactly known. That is why we recommended intensive studies about specificity, clearance time… about CCA before using POC-CCA alone for assessment of cure.

21.Insert if any limitation

Response: we have put limitations of the study in the last paragraph of the ‘discussion’ section

22.The conclusion might be okay if you recommend continuous study at a large area and the country level.

Response: we accept the comment and revised the conclusion accordingly (last sentence of the ‘conclusion’ section).

Reviewer #3

General comment

The authors of the current study are working on a well-known issue. The data could be useful for Ethiopia. Here I have some comments and suggestion for improvement.

Title

= “Efficacy and safety of praziquantel for the treatment of Schistosoma mansoni infection across different transmission settings in Amhara Regional State, northwest Ethiopia” This title should be revised to consider the content of the MS. Note sure if this study was implemented in several transmission settings. What is clear is that the study was targeting SAC with different level of infection intensity (low, moderate and heavy)

Response: we have re-analyzed the efficacy based on endemicity level of data collection districts (low, moderate, high endemic) in the revised manuscript and we believe now the title is in line with the contents of the manuscript (please see revisions in the ‘study area’, ‘sampling technique’ and ‘discussion’ sections, and Table 3)

Abstract

= “Factors such as reduced PZQ efficacy after … contributing to persistent transmission of the disease.” Regarding the context in Ethiopia, this could not the rationale behind this study. The authors should find good rationale supporting the current work

Response: revised (please see the highlighted in the ‘background’ section of the abstract)

Introduction

= “More than 240 million people are infected globally” This statement is not anymore true

Response: we have got an updated data from global atlas of ‘helminth infections’ it to be more than 200 million. Here is the link of the reference https://www.thiswormyworld.org/worms/global-burden

= “Reduced PZQ efficacy following multiple rounds of MDA might have contributed for the continued incidence and this has been evidenced by recent studies.” As mention above the rationale behind this study must be reconsidered.

Response: we have made revisions (highlighted in the background section)

Materials and methods

= “From February to June 2023, a school-based single-arm …….different transmission settings of intestinal schistosomiasis” Which are these different settings? How those settings have been selected?

Response: transmission settings are defined in the revised manuscript and how data collection areas representing each setting were selected is also explained (please see the highlighted in the ‘study area’ and ‘sampling technique’ sections)

= “the calculated sample size was 635. For the PZQ efficacy study, we selected the above mentioned five schools to which a total of 283 participants were allocated.” What is the precise sample size of this work? How precisely this sample size has been estimated? Please, generate a new section entitled “Sample size calculation” and describe clearly how the sample size has been assessed.

Response: thank you for this critical comment. we have added a new section ‘sample size calculation and sampling technique’ and have shown in more detail about the sample size calculation and sampling techniques

= Please, define clearly your primary outcome and secondary outcome and show us how this has been considered in your sample size calculation.

Response: we have added a new sub-heading (study outcomes and definitions) under the ‘methods’ heading in the revised manuscript. As explained there, the primary outcome is PZQ efficacy (CR and ERR) while occurrence of adverse events is a secondary outcome. We calculated the sample size considering the primary outcome following the WHO protocol for assessing anti-helminthic drug efficacy. 

= L151-L161 “Urine POC-CCA test” this part could be shorten avoiding speculation

Response: Revised based on the comment 

= “Duplicate KK slides were prepared and examined from each sample and the average egg count was taken as the final EPG” This statement is not correct. Please, revise since 24 to get the EPG should multiply the average egg count from both slides.

Response: Revised based on the comment

=”Schoolchildren who lived in the study area at least for six months, attended a regular primary school, gave consent/assent to participate in the study were included.” SAC could not consent they are only assenting and their parents are consenting.

Response: revised based on the comment

=”Evaluation of praziquantel efficacy” Please specify the brand of the praziquantel used in the current study

Response: specified in the revised manuscript (brand name: biltricide, manufacturer: Bayer pharmaceuticals)

= L188 – L190 “Treated children were then counseled to avoid activities related to S. mansoni infection …. in a pot/jar for ≥24 hours before use).” Remove this part since praziquantel is not active against juvenile worms. So, whatever the precautions you are taking to avoid reinfection you could not clear the juvenile work in the children body. The follow up time set to three weeks has been set to limit the impact of reinfection on cure rate and ERR calculation.

Response: we have removed this part as per your recommendation

= How many time the adverse events have been follow?

Response: occurrence of adverse events was expected within one week after treatment and treated children were followed on a daily basis from day 1 to day 7. However, parents/caregivers were communicated to call on phone and communicate with the study team in case any health complaint was there until day 21 after treatment (revision highlighted).

= L214: “STH infected vs non-infected” This is the first time STH are mentioned. Nowhere, in the previous section something is said about STH.

Response: STHs (A. lumbricoides, hookworms and T. trichuria) are co-endemic with S. mansoni in Ethiopia and hence PZQ and albendazole/mebendazole are co-administered to school-aged children during MDA campaigns. Hence we were interested to assess if there is a difference in CR and ERR among groups who were infected with S. mansoni alone (treated with PZQ) and groups who were co-infected (treated with PZQ plus albendazole). We have added explanations about STHs in the ‘background’ and ‘methods’ sections.

= L214: “while Kruskal-Wallis H test was used to compare the mean EPG among participants with light, moderate and heavy intensity of infections” Why this comparison since the groups are already well defined as low, moderate and heavy infected participants?

Response: our main intension here was not to compare baseline EPG which was used to define participants as with light, moderate and heavy infections. But we were interested to compare the post-treatment EPG between participant groups who were with light, moderate and heavy infection at baseline. i.e. to check whether fecal egg count reduction is proportional to the baseline infection intensity or not. 

= L234: Please write “Cure rate” instead of “Cure rate and associated risk factors”

Response: revised as per your comment

Results

= Figure 1: should be revised to make it understandable including a figure caption

Response: the handling editor has raised a copyright issue and recommended to remove this figure. Therefore, we have removed this figure from the revised manuscript submission

= Table 4: Please, could you provide confidence interval for the ERR

Response: we have added the 95%CI for the ERR in the table. However, due to a recommendation from reviewer 1, we have attached this table as a supplementary file (Supporting information S2) in the revised submission rather than including it in the body of the manuscript

= Table 5 should be revised as follow:

Variable

Category Number of children assessed Adverse event

n (%)

χ2

P-value

Response: revised based on your suggestion (highlighted in the revised manuscript)

Discussion

= “The CR among SC with light infection (95.7%) by KK was higher than the CR among SC with moderate (81.5%) and heavy (64.3%) infections (χ2 = 12.53, p = 0.002)” One issue here is also the insensitivity of the KK in low infected people and need to be discussed.

Response: we accept the comment and we have added in the revised manuscript about the role of KK’s low sensitivity in falsely raising CR among lightly infected participants. 

= Please, make you discussion around your key findings by avoiding speculation

Response: we have made an intense revision of the ‘discussion’ section and have removed speculations and explanations which are not directly related to our study objective and findings. 

= Due to the limited sample size in the different study groups, the results of the current study should be interpreted with caution.

Response: we agree and we have explained this in the ‘discussion’ and ‘conclusions’ section in the revised manuscript

---

## [Decision Letter · Decision Letter 1]

16 Jan 2024

PONE-D-23-33684R1Efficacy and safety of praziquantel for the treatment of Schistosoma mansoni infection across different transmission settings in Amhara Regional State, northwest EthiopiaPLOS ONE

Dear Dr. Abebe,

Thank you for submitting your manuscript to PLOS ONE. After careful consideration, we feel that it has merit but does not fully meet PLOS ONE’s publication criteria as it currently stands. Therefore, we invite you to submit a revised version of the manuscript that addresses the points raised during the review process.

We look forward to receiving your revised manuscript.

Kind regards,

Clement Ameh Yaro, Ph.D

Academic Editor

PLOS ONE

Journal Requirements:

Reviewers' comments:

Reviewer's Responses to Questions

**Comments to the Author**

1. If the authors have adequately addressed your comments raised in a previous round of review and you feel that this manuscript is now acceptable for publication, you may indicate that here to bypass the “Comments to the Author” section, enter your conflict of interest statement in the “Confidential to Editor” section, and submit your "Accept" recommendation.

Reviewer #1: All comments have been addressed

Reviewer #2: All comments have been addressed

2. Is the manuscript technically sound, and do the data support the conclusions?

Reviewer #1: Yes

Reviewer #2: Yes

3. Has the statistical analysis been performed appropriately and rigorously? 

Reviewer #1: Yes

Reviewer #2: Yes

4. Have the authors made all data underlying the findings in their manuscript fully available?

Reviewer #1: Yes

Reviewer #2: Yes

5. Is the manuscript presented in an intelligible fashion and written in standard English?

Reviewer #1: Yes

Reviewer #2: Yes

6. Review Comments to the Author

Reviewer #1: The manuscript has been greatly improved after revision. Some minor comments and questions from my side.

Abstract: nothing mentioned about efficacy/safety in different transmission settings. Since this is something that makes this manuscript unique, I think it would be good to give it some attention in the abstract.

Line 114-116: here ‘in different transmission settings’ should be added to the objective of the study. ‘Hence, the present study aimed to assess PZQ efficacy and safety for the treatment of S. mansoni infection and to compare the capacity of POC-CCA and KK for assessing cure in different transmission settings’.

Sample size calculation:

Prevalences are based on microscopy I believe, this should be added to the text to be clear since you will be comparing microscopy with POC-CCA.

I do not understand the 29% prevalence in low endemic area, how can this be so ‘high’? I think the authors are trying to explain this, but it remains unclear to me.

I would suggest to make a new paragraph of ‘Sampling technique’

Table 4. Number of children assessed, the percentage should be of the subgroup, not of the total group. For example, now it looks like only 12% of the 6-9 year olds were examined for adverse events, while all of them were actually examined. Unless in some subgroups not all children were assessed, I do not see a reason why this column should be added to the table.

What was the distribution of low/moderate/heavy intensity infections among the 3 endemicity districts? I know the numbers might be small, but I would be interested to see this. For example, do all the heavy intensity infections come from the high endemic area? Or do you also find them in the low endemic area? It would be nice to add this information to the first table, or as supplementary material or even added in a few lines in the text. I think this is very informative.

Reviewer #2: This study has made appropriate corrections during the review process, therefore, I suggest that it be accepted.

7. PLOS authors have the option to publish the peer review history of their article (what does this mean?). If published, this will include your full peer review and any attached files.

Reviewer #1: **Yes: **Pytsje T Hoekstra

Reviewer #2: No

---

## [Author Response · Author response to Decision Letter 1]

17 Jan 2024

Dear editor and reviewers, thank you for your constructive comments again which all are important inputs for the betterment of the manuscript. Below we tried to respond to all the comments/questions one by one; we also have incorporated all the corrections in the revised manuscript (shown as highlighted). 

Editor 

Please review your reference list to ensure that it is complete and correct. If you have cited papers that have been retracted, please include the rationale for doing so in the manuscript text, or remove these references and replace them with relevant current references.

Response: we have re-checked all the references and we have confirmed all references in the ‘references’ list are correctly cited in the body of the manuscript; and all cited references are included in the list. We have also ensured that the references and citations are made based on the journal guideline. We did not cite any paper that has been retracted.

Reviewer #1: 

Abstract: nothing mentioned about efficacy/safety in different transmission settings. Since this is something that makes this manuscript unique, I think it would be good to give it some attention in the abstract.

Response: comment accepted and we have added results about CR and adverse events across the three transmission settings (highlighted). We have also added data about adverse events in different transmission settings in Table 4. 

Line 114-116: here ‘in different transmission settings’ should be added to the objective of the study. ‘Hence, the present study aimed to assess PZQ efficacy and safety for the treatment of S. mansoni infection and to compare the capacity of POC-CCA and KK for assessing cure in different transmission settings’.

Response: we accept the comment and have amended the objective accordingly. 

Sample size calculation:

Prevalences are based on microscopy I believe, this should be added to the text to be clear since you will be comparing microscopy with POC-CCA.

Response: we accept the comment and we have explained this in the revised manuscript.

I do not understand the 29% prevalence in low endemic area, how can this be so ‘high’? I think the authors are trying to explain this, but it remains unclear to me.

Response: we expect it will be confusing for readers (that is why we tried to explain in the manuscript). In Ethiopia, Schistosomiasis mapping was done at a district level. They took samples from schools a district and the endemicity level of the district was determined based on the mean prevalence from schools. However, there might be great variations across schools or villages within the same district. So, if researchers recruit participants from a school/village where there is high Schistosoma focal transmission, the prevalence might be higher than the respective district’s average prevalence. 

 I would suggest to make a new paragraph of ‘Sampling technique’

Response: comment accepted and we have revised the manuscript accordingly

Table 4. Number of children assessed, the percentage should be of the subgroup, not of the total group. For example, now it looks like only 12% of the 6-9 year olds were examined for adverse events, while all of them were actually examined. Unless in some subgroups not all children were assessed, I do not see a reason why this column should be added to the table.

Response: it was just to give information about the proportion of each group among the total. Now we ensured that such information is presented in Table 1 and we have removed the percentage in Table 4, to avoid confusions. 

What was the distribution of low/moderate/heavy intensity infections among the 3 endemicity districts? I know the numbers might be small, but I would be interested to see this. For example, do all the heavy intensity infections come from the high endemic area? Or do you also find them in the low endemic area? It would be nice to add this information to the first table, or as supplementary material or even added in a few lines in the text. I think this is very informative.

Response: Thank you. We have presented this data as supporting information (second table in Supporting information S1) in the revised submission. We have also explained this in the ‘results’ section.

Reviewer #2: 

This study has made appropriate corrections during the review process, therefore, I suggest that it be accepted.

Thank you

---

## [Decision Letter · Decision Letter 2]

23 Jan 2024

Efficacy and safety of praziquantel for the treatment of Schistosoma mansoni infection across different transmission settings in Amhara Regional State, northwest Ethiopia

PONE-D-23-33684R2

Dear Dr. Abebe,

We’re pleased to inform you that your manuscript has been judged scientifically suitable for publication and will be formally accepted for publication once it meets all outstanding technical requirements.

Kind regards,

Clement Ameh Yaro, Ph.D

Academic Editor

PLOS ONE

Additional Editor Comments (optional):

Reviewers' comments:

Reviewer's Responses to Questions

**Comments to the Author**

1. If the authors have adequately addressed your comments raised in a previous round of review and you feel that this manuscript is now acceptable for publication, you may indicate that here to bypass the “Comments to the Author” section, enter your conflict of interest statement in the “Confidential to Editor” section, and submit your "Accept" recommendation.

Reviewer #1: All comments have been addressed

2. Is the manuscript technically sound, and do the data support the conclusions?

Reviewer #1: Yes

3. Has the statistical analysis been performed appropriately and rigorously? 

Reviewer #1: Yes

4. Have the authors made all data underlying the findings in their manuscript fully available?

Reviewer #1: Yes

5. Is the manuscript presented in an intelligible fashion and written in standard English?

Reviewer #1: Yes

6. Review Comments to the Author

Reviewer #1: (No Response)

7. PLOS authors have the option to publish the peer review history of their article (what does this mean?). If published, this will include your full peer review and any attached files.

Reviewer #1: **Yes: **PT Hoekstra

---

## [Editor Report · Acceptance letter]

23 Feb 2024

PONE-D-23-33684R2 

PLOS ONE

Dear Dr. Abebe, 

I'm pleased to inform you that your manuscript has been deemed suitable for publication in PLOS ONE. Congratulations! Your manuscript is now being handed over to our production team.

Kind regards, 

on behalf of

Dr. Clement Ameh Yaro 

Academic Editor

PLOS ONE